### 1 Seasonal forecast verification and application in times of change

- 2 Yoav Levi<sup>1</sup> Itzhak Carmona<sup>1</sup>
- <sup>3</sup> <sup>1</sup>Israel Meteorological Service, Bet-Dagan, 50250 Israel
- 4 *Correspondence to*: Yoav Levi (leviyo@ims.gov.il)
- 5

6 Abstract. Seasonal forecast is being promoted as one of the climate services given to the public and decision 7 makers also in the extra-tropics. However seasonal forecast is a scientific challenge. Rapid changes in climate and the socio-8 economic environment in the past 30 years introduce even a bigger challenge for the end-users of seasonal forecasts based 9 on the past 30 years.

10 Decision makers should relay on a forecast only if they fully understand the forecast skill and the forecast will not 11 be a completely erroneous. Therefore, the percentage of forecasts for above normal condition that realized to be below 12 normal conditions and vice versa is measured straightforwardly by the "Fiasco score". To overcome the climate and socio-13 economicenvironment changes an attempt to relate the next seasonal forecast to the previous season forecast and observed 14 valueswas tested. The findings indicate that ECMWF system-4 seasonal forecast skill for June-July-August (JJA) 15 temperatures for the marine tropics is very promising as indicated by all the skill scores, including using the previous JJA 16 forecast as the base for the next JJA season. However for the boreal summer temperatures forecastover land, the main source 17 of the model predictability originates from the warming trend along the hindcastperiod. Over the Middle East and Mongolia 18 removing the temperature trend eliminated the high forecast skill. Evaluation of the ability of the next season forecast to 19 predict the changes relative to the previous year's season has shown a positive skill in some areas compared to the traditional 20 30 years based climatology after both forecasts and observed data were de-trend.

22

#### 23 **1. Introduction**

24 Due to the chaotic nature of the atmospheric circulation, which is ostensibly non-periodic, prediction of the sufficiently 25 distant future is impossible by any method unless the present conditions are known exactly. In view of the inevitable 26 inaccuracy and incompleteness of weather observations, precise very Long Range Forecast (LRF) would seem to be non-27 existent (Lorenz 1963). However despite the chaotic nature of the atmosphere the lower-boundary forcing, which evolves on 28 a slower time-scale than that of weather, can impart significant predictability on atmospheric development (Palmer and 29 Anderson 1994). Ensemble prediction systems provide the means to estimate the flow-dependent growth of uncertainty 30 during a forecast. Multi-model and related ensembles are vastly superior to corresponding single-model ensembles, but do 31 not provide a comprehensive representation of model uncertainty. (Palmeret al. 2005).

32 Changes in sea surface temperature (SST) are the major drivers of seasonal forecast. The El Niño-Southern 33 Oscillation (ENSO) is the leading mode of inter-annual variability, with global impacts on weather and climate that have 34 seasonal predictability (Hoell et al. 2014). The linear nature of tropical dynamics andnear surface winds which are strongly 35 constrained by the ocean (Lindzen and Nigam 1987) are the source of the tropical areas predictability. However, the inter-36 annual variability of tropical SST outside of the central and eastern Pacific is small and less predictable (Barnston et al. 37 2010). In the extra-tropics winds are poorly constrained by the ocean and then predictability is even lower (Smith et al. 38 2012). Nevertheless, there are evidences for extra-tropics predictability. The predictions for precipitation of the southern part 39 of United States, derived by ENSO and Pacific Decadal Oscillation (PDO) had a success rate of almost 77% (Kurtzman and 40 Scanlon 2007). Eruptions of volcanoes, solar radiation, Atlantic multi-decadal variability (AMV), snow cover, soil wetness 41 and the quasi-biennial oscillation (QBO) have been shown to be sources of extra-tropics positive seasonal forecast skill 42 (Folland et al. 2012, Smith et al. 2012, Barnston et al. 2010).

LRF validation is done by verifying the ability of the model ensemble to reforecast (hindcast) the past climate and to determine whether the model ensemble is capable of following the observed inter-annual variability. A common method for presenting seasonal forecast is to divide both observes and forecast distributions to three equal probability

terciles(Barnston et al. 2010). A deterministic forecast will use the ensemble mean or median to determine the expected
tercile. A probabilistic forecast will assign the probability of each tercile-based category.

There are several methods for evaluating LRF hindcast skill: Simple Pearson correlations coefficients for deterministic forecast (Hoell et al. 2014, DelSole and Shukla 2010, Kim et al, 2012), the Area Under the Relative Operating Characteristic (AUROC) curve (Mason and Graham 2002, Fawcett 2008, Kharin and Zwiers, 2003) which measures the hit rate (HR) vs. the false alarm rate (FAR) and the Ranked Probability Skill Score (RPSS) which measures the accuracy of probabilistic forecasts (Kumar et al. 2001). These methods are in common practice in the scientific community and each of them has its strength and weakness.

54

55 The end-user which needs to take action in view of seasonal forecasts should consider the risks and the benefit-cost 56 ratio of his actions. Our main goal is to evaluate the seasonal forecast taking into account the rapid changes in both climate 57 and socio-economic development. For many end-users the deviation from 1981-2010 average condition may not be useful as 58 their working environment may have changed dramatically during this period. Local stakeholders planning adaptation 59 measures need to understand the effect of their environmental changes. In hydrology there is a large impact of land use 60 changes as urbanization and vegetation on watershed stream flow (Ohana-Levi et al. 2015). For drought planning growing population and standard of living leads to increase of water consumption (Wilhite 2012). The vulnerability of populations to 61 62 heat wave is changing as population acclimatizes by using air-condition (Lundgren et al. 2013). In agriculture crop yields 63 change as fertilizers and pesticides are penetrating (Matson et al. 1997) together with the increase of CO<sub>2</sub> concentrations and 64 climate trends themselves (Lobell and Fields 2007). The dairy industry has changed dramatically as milk production per 65 cow increased (Lucy 2001).

Furthermore Folland et al. (2012) showed that the impact of temperature trends on the seasonal forecast skill in northern Europe was the most significant predictor compared to ENSO, volcanoes, NAO and QBO. Therefore, a simple enduser does not have the ability to estimate the impact of the coming season forecasted climate relative to the past 30 years. However end-users do remember both the last year climate and their figures concerning their work. If the seasonal forecast

# 

- will give the change in climate relatively to the previous year season, the stakeholders could plan to take action to mitigate
  the impact of above or below normal conditions relative to the previous year.
  Our main goal is to find a simple method for the end-user to use the ECMWF system 4 (Sys4) seasonal forecasts
  and to understand the skill of the forecast in order to assess the risks and perform cost-benefit analysis of using the forecast.
  The goal will be achieved by verifying the global Sys4 seasonal re-forecasts (Molteniet at. 2011) for June-July-August (JJA)
  temperature against ECMWF ERA-Interim reanalysis.
- 76

### 77 **2. Model data**

### 78 2.2 ECMWF system 4 system

The ECMWF Sys4 is a coupled ocean-atmosphere dynamical model with a horizontal resolution of ~0.7° and 91 vertical levels (T255L91). It became operational at November 2011 with 51 ensemble members. The hindcast was performed for 30 years from 1981 to 2010 but only with 15 ensemble members created by SST perturbations and the activation of stochastic physics. Therefore, a total number of 450 runs are available to construct 30 years of model climatology and its verification (Molteni et al. 2011). The current work is done with one moth lead forecasts for JJA.

### 84 2.2 ERA-Interim reanalysis

Reanalysis is only an estimation of the climate status and it is not even purely homogenized with time. Nevertheless In order to verify the Sys4 forecasts the ECMWF ERA-Interim reanalysis was chosen, as a huge number of observations are assimilated to the model. The number of assimilated observations increased from approximately 10<sup>6</sup> per day on average in 1989, to nearly 10<sup>7</sup> per day in 2010 (Dee at al. 2011). Furthermore, the ERA-Interim model has the same spatial resolution as the ECMWF Sys4 model. Therefore, all the grid points of the hindcast (512\*256=131,072) were verified with the same ERA-Interim grid point. As the data quality near the polar areas is less reliable (Dee at al. 2011) it is not presented in the maps.

#### 93 **3. The "Fiasco score", AUROC and RPSS**

For tercile forecasts there are 9 possible outcomes events in the forecasted vs. observed contingency table each containing an equal probability of 11.11%. When the seasonal forecast ensemble median resides in the observed tercile the deterministic forecast is counted as a correct forecast (hit). If the ensemble median does not reside in the observed tercile it is regarded as false forecast. A complete failure forecast (a Fiasco) occurs if a forecast for above normal condition is materialized to be below normal conditions or vice versa. The "Fiasco score" evaluates the fiasco percent of cases where two categories reside between observed and forecasted. By random probability, the chance for a hit is 33.3%, the chance for a false forecasted season is 44.4% and the chance for a Fiasco forecast is 22.2%.

# 101 The AUROC score (Kharin and Zwiers, 2003) which is used to evaluate above or below normal conditions is based 102 on the hit rate (HR) and the false alarm rate (FAR) as defined:

103

$$HR = \frac{H}{O}$$
 and  $FAR = \frac{FA}{NO}$  (1) 104

105

- Where: H is the number of hits (events forecasted and occurred); O is the number of events that Occurred; FA is the
  number of false alarms (events were forecasted but did not occur); NO are the number of events which did not Occur.
- 108

109 As the seasonal forecast is given for 3 categories with equal random probability the observed and not observed 110 events are constant. For a hindcast period of 30 years there are always 10 events above normal and 10 events below normal 111 conditions (0 = 10). Therefore, always 20 events do not occur (NO = 20).

112

For a given probabilistic forecast–observation pair, the Ranked Probability Score (RPS) is defined for 3 categories as:

116 
$$RPS = \sum_{m=1}^{3} (F_m - O_m)^2 (2)$$

# 

Where F<sub>m</sub> and O<sub>m</sub> denote the m<sup>th</sup> component of the cumulative forecast and observation vectors F and O, 117 118 respectively. 119 120 The ranked probability score is essentially an extension to many-event situation of the Brier score which is a mean 121 squared-error score for verification of probabilistic forecasts of dichotomous events. The observation is assigned with 1 if the 122 forecast event occurs and 0 if the event does not occur. The ranked probability skill score (RPSS) relates RPS and RPS<sub>clim</sub> 123 which is the value expected by climatology where each category has equal probability (Wilks 2006) by eq. (3). 124  $RPSS = 1 - \frac{\langle RPS \rangle}{\langle RPS_{clim} \rangle}$ (3) 125 126 127 The RPSS measures the forecast whole distribution (all 9 possible outcomes events) including the around normal 128 cases which are ignored by the AUROC and the "Fiasco score". The above and below normal AUROC takes into account 6 129 out of 9 possible outcomes events compared to 2possible outcomes events evaluated by the "Fiasco score". However, despite the low robustness of the "Fiasco score" it has a strong correlation with the above and below normal AUROC (r = -0.87), and 130 131 RPSS (r = -0.67) calculated for 131,072 global points. Spatial averaging of the model skill increase the robustness of the 132 "Fiasco score" as it reduces possible sampling errors and uncertainty noise of a single measure. Figure 1 presents the latitude 133 averages of the AUROC and RPSS as a function of the latitude average of the simple "Fiasco score". High correlations 134 coefficient with the AUROC (r= -0.99) and RPSS (r = -0.88) indicate that the "Fiasco score" may serve as an additional 135 simple and straightforward score for end-users to assess seasonal forecast for their needs.

136 It can be seen in Figure 1 that all forecast skills increases equator ward. If the AUROC score is above 0.5 and the 137 "Fiasco scores" is lower than 22% the forecast is skilful (compared to random variability). Therefore, all latitude average 138 points in Figure 1 have a positive skill. However the latitude averages RPSS, which measures more rigorously the whole 139 forecast ensemble skill, are positive only in the tropic (latitudes  $

#### 141 **4. Temperature trends between 1981 and 2010**

142 Figure 2a presents the ERA-Interim JJA 2 m temperature trend between 1981 and 2010 indicated by the linear regression slope. The highest warming rates, above 1°C decade<sup>-1</sup>, are observed in the Middle East, Mongolia and the Labrador Sea. The 143 144 map in Figure 2a is broadly consistent with NOAA's Merged Land-Ocean Surface Temperature (MLOST) analysis (Vose et 145 al. 2012). The main differences between these two analyses are in Scandinavia, Central Asia and South Africa, where the 146 ERA-Interim trends are moderate compared to MLSOT. Figure 2b presents the Sys4 temperature trend with the same scale of (Figure 2a. The global ERA-Interim averages trend between 60°S and 70°N is 0.13°C decade<sup>-1</sup> which is similar to the Sys4 147 global average trend which is  $0.14^{\circ}$ C decade<sup>-1</sup>. Although both average trends are not significantly different (P-value < 0.05), 148 149 it is evident that the two maps are substantially different. On the one hand over the oceans Sys4 trend are positively biased 150 compared to the reanalysis ((Figure 2c). On the other hand Sys4 trends are strongly negatively biased were the reanalysis 151 indicates very strong warming as the Middle East and Mongolia. For example, in Iran where the ERA-Interim trend reaches 152 a value of 1.4°C decade<sup>-1</sup> the Sys4 trend is only 0.3°C decade<sup>-1</sup>.

For the evaluation of the hindcast period these trend differences are not a problem as the terciles are calculated for each series separately. However if both Sys4 and ERA-Interim trends will maintain in the future, most years will be correctly forecasted as above normal compared to the 1981-2010 conditions. Furthermore, these differences may influence the lowerboundary forcing which is the source of seasonal forecast predictability.

157

#### 5. The "Fiasco next score" forecast skill from one year to the next.

Seasonal forecasts use 30 year reference periods to determine the seasonal forecast conditions. If during this period there are temperature trends together with socio-economic and environmental changes the usefulness of the seasonal forecast for the stakeholders may be questioned. To eliminate these changes an attempt to use the previous year's season condition as a reference for the next season was tested. In order to assess the forecast skill, one year leg differences of the forecast and observed are examined. The forecasted and observed differences are divided into 3 equal probability groups to define the normal, above and below normal conditions. For a hidcast period of 30 years only 29 differences between previous and next season are available. Therefore each case has a probability of 3.45% instead of 3.33% for the 30 year reference period.

# 

Figure 3 presents the latitude averages of the "Fiasco next score", with the previous year season serving as the reference for the next season, as a function of the 30 years "Fiasco score". In the tropic  $20^{\circ}$ S –  $20^{\circ}$ N where the JJA temperatures are not changing much ((Figure 2) the average decreases in skill is 1.7% meaning an addition of one forecast failure in 60 years. In the mid-latitudes there is a significant (p-value <0.05) difference between the 2 hemispheres. In the boreal summer the forecasts skill deterioration relative to the "fiasco score" is double compared to the southern hemisphere winter skill.

Most end-users are interested in forecast over land. Therefore, an attempt to compare over land the "Fiasco next score" to the de-trended "Fiasco score", obtained by de-trending boththe forecast and observed (reanalysis) 30 year of data, is presented in Figure 4. As there are substantial temperature trends in the past 30 years especially over land (Figure 2) this attempt reduced the global average difference to less than 2%. In the tropic  $20^{\circ}$ S –  $20^{\circ}$ N there is no significant difference between the two average skill scores (p-value > 0.05). Furthermore in the northern hemisphere tropics the "Fiasco next score" is significantly better by 0.8% compared to the "Fiasco score".

177

Figure 5a presents the global JJA 2 m temperature hindcast skill evaluated by the "Fiasco score" based on the 1981-2010 reference period. It can be seen that the tropic Pacific Ocean is the largest area with high predictability, indicated by the absence of cases where the model failed to distinguish between above and below normal conditions. At the same time, there are also areas in the extra-tropics as the Labrador Sea near Greenland, Bering Sea, Gulf of Alaska, the Middle-East and Mongolia where the "Fiasco scores" approaches zero indicating high skill to distinguish between above and below normal conditions. It is also evident that there are large regions in the tropics such as tropical Africa and Brazil where the "Fiasco score" approaches the no skill level of 22%.

Figure 5b presents an exercise to de-trend linearly both forecasted and reanalysis datasets. The most prominent effect of the de-trending on the forecast skill occurs in the boreal summer over land between 20°N and 60°N, where the average temperature trends reached 0.38°C per decade. In areas with strong warming trends as Mongolia, Europe and the Middle East, where the warming rate reaches 0.48°C per decade, the de-trending was detrimental for the forecast skill as the "Fiasco score" more than doubled, growing from 6.4% to 14.5%. In Central Asia where weak cooling trends were observed

# 

in 40% of the area, the de-trending improved the forecast skill by a factor of two, however the overall skill remained verylow.

192 Figure 5c presents the global JJA 2 m temperature hindcast skill evaluated by the "Fiasco next score" where the next 193 season forecast is given relative to the previous year's season. The global average "Fiasco next score" is higher by 3.4% 194 compared to regular "Fiasco score", indicating that the price for using the previous season as a reference is an increase of one 195 complete failure forecast in 30 years. However compared to the de-trend "Fiasco score" the global "Fiasco next score" is 196 higher only by 1.6% reducing the price to only one additional complete failure forecast in 60 years. From Figure 5 it is 197 evident that in the continental areas of the Middle East and Mongolia the high forecast skill ((Figure 5a) disappeared after 198 de-trending ((Figure 5b) and the "Fiasco next score" (Figure 5c) are close to a radon probability forecast of 22%. However in 199 the Labrador Sea and the Gulf of Alaska the forecast remains skilful although an increase of 1 or 2 fiasco cases in 30 years 200 (3.4-6.7%) is evident. In tropical Africa between 10°S to 10°N the difference between the "Fiasco score" and the "Fiasco 201 next score" is not significantly different (p-value > 0.05). In Nigeria and Southern Chad the "Fiasco next score" is even 202 significantly lower compared to the 30 years reference "Fiasco score" after de-trending. It is also evident that there are large 203 regions where the next year method has a significant (p-value 

### 

| 215 | At the equator there is a prominent reduction in both reanalysis temperature warming trend and model skill indicted       |
|-----|---------------------------------------------------------------------------------------------------------------------------|
| 216 | by all scores (also AUROC which is not presented). The most significant minimum is of the RPSS which is evident exactly   |
| 217 | on the equator. The fact that warming trends and forecast skill reaches minimum values exactly at the equator may suggest |
| 218 | that it is associated to a dynamic effect linked to the Coriolis force which is zero on the equator. Explanations such as |
| 219 | Equatorial Kelvin Wave, Equatorial Divergence or Equatorial Undercurrent (EUC) are beyond the scope of this paper.        |

### 220 **7. Conclusions**

The aim of this work is to help end-users to understand better how to use seasonal forecasts. The end-user should determine whether the benefits of taking action in view of the available seasonal forecast, outweigh the costs of ignoring the forecast. It is clear that in case a forecast for above average condition is materialized to become below average conditions or vice versa the overall use of seasonal forecast will cause more damage that benefit.

The evaluation of the Sys4 seasonal forecast hindcast for JJA temperature shows thatthe whole forecast probability is skilful only in the tropics as indicated by the RPSS (Figs. 1, 6). However, the Sys4 skill to distinguish between the upper most and lower most parts of the observed distribution is positive also in extra tropicsareas as indicated by both the AUROC and "Fiasco score". It is evident that a large component of JJA temperature forecasts skill for the boreal summer over land (as the Middle East and Mongolia) originated from the temperature trends in the hindcast period (Figs. 4, 5, 6).

The spatial average of the simple and intuitive "Fiasco score" is highly correlated to the AUROC curve (r = -0.97) and to the RPSS (r = -0.87) and can be used by end-users to identify whether the hindcast is capable to distinguish between the upper most and lower most parts of the observed distribution ((Figure1). Using such a deterministic approach is in line with Chen and Kumar (2015) finding that there are small systematic year to year variations in the ensemble probability density function (PDF) spread. They suggested that it might be a good practice in seasonal predictions to assume that the spread of seasonal means from year to year is constant and the skill in seasonal forecast information resides primarily in the shift of the first moment of the seasonal mean of the PDF.

In order to minimize both climate trends ((Figure2) and the changing factors of end-users practice such as crop management, population growth or socio-economic development, using the previous year's season as a reference for the next

season forecast is suggested. It is shown that for limited areas like Nigeria and Southern Chad, between Australia and New Caledonia and to east of the Philippines the "Fiasco next score" over performs the "Fiasco score" before and after detraining. This extreme solution is obviously not suggested to replace the robust traditional 30 year reference period which is shown to over perform the average for most of the globe. However the end-user should consider using the coming season forecast relative to previous season or a shorter reference period that the traditional 30 years in times when both climate and his practice are undergoing rapid changes.

It is encouraging to find that over the Labrador Sea, where very high temperature trends were observed (Figure 2a) and large amounts of heat is releases to the atmosphere (Lazier et al. 2002), de-trending did not eliminate the seasonal forecast predictability (Figure 5b). In line also the "Fiasco next score" indicates that the Sys4 remains skilful (Figure5c). This fact emphasizes that the source of predictability lays in the oceans also in the extra-tropics. As the SyS4 skill for Iceland and the Azores Islands areas is relatively low it would be suggested to find a predictor based on the Labrador bay area to enhance seasonal teleconnection as the summer North Atlantic Oscillation (NAO).

Further study is needed to understand the minimal values, exactly at the equator, of both the ERA-Interim reanalysis warming trend and the forecast skill. Li and Xie (2014) showed that the excessive equatorial Pacific cold tongue bias and double Inter-Tropical Convergence Zone (ITCZ) stand out as the most prominent errors of the ocean current generation of coupled general circulation models (CGCMs).