# Peer review of "Seasonal forecast verification and application in times of change"

_Earth System Dynamics, 2016_

## Referee Comment (RC1) · Anonymous Referee #1 · 31 Dec 2016

To facilitate the assessment of risks, seasonal forecasts are usually expressed as the deviation relative to a 30-yr average condition for end-users. However, the climate and socio-economic changes during this period may enable the deviation questionable. To eliminate these changes, this paper introduces a method, which uses the previous year's season condition as a reference for the next season, to evaluate the skill and usefulness of the ECMWF system 4 (Sys4) seasonal June-July-August (JJA) temperature forecasts, wherein the ECMWF ERA-Interim reanalysis is employed as the surrogate observations. The results may be valuable for the policy-makers who are interested in the application of seasonal forecasts. The paper is organized logically and well writing. However, I still have a few concerns that need to be addressed.

Major comments:

[Figure]

1. Policy-makers or end-users may have great interests on the impacts of extreme event to the socio-economy, agriculture, and environment. In this paper, however, the metric of "Fiasco score" was constructed based on the above-normal and below-normal condition which does not simply equal to extreme event. The skill of Sys4 in terms of capture the extremes event should be added or discussed in the context.

2. As Figure 6 shows, the forecast skill derived by the new method (i.e., using the previous year's season as a reference for the next season forecast) tends to be lower than that derived from the traditional 30 year reference period. As well, the authors concluded that the new method is obviously not suggested to replace the robust traditional 30 year reference period (see Line 241-242). If so, why you attempt to use it?

3. In my view, the precipitation or streamflow forecasts, as compared with temperature, may have more significance on end-users practice. Why you put the primary attention just on temperature, rather than precipitation?

Specific comments:

1. Section 2 – Model data. Please provide the archive website for ECMWF system 4 system data and ERA-Interim reanalysis, respectively.

2. Line 94-95. Please add an example table to explicitly illustrate the contingency table, so that the readers are more easily to understand it.

3. Line 101-102. Please provide more details (e.g., equations) for the calculation of AUROC.

---

## Referee Comment (RC2) · Anonymous Referee #2 · 9 Feb 2017

Summary

In a context of developing seasonal forecasts as climate services, this paper proposes an evaluation of seasonal temperature forecasts from ECMWF System 4 for the June-July-August season. Authors first propose the Fiasco score to evaluate the forecasts for risk assessment and cost-benefit analysis and relate it to the common RPSS and AUROC scores. They also present the trends in temperature from System 4 and ERA-interim. They conclude that, with such trends, classifying forecasts between above-normal, normal or below-normal conditions based on long-term thresholds will not be useful for end users as all future forecasts will be above-normal. If I understand correctly, the authors then analyze the System 4 temperature forecasts, along with two light post-processing: the detrended System 4 temperature forecasts and the series of differences in temperature forecasts between the current season and the previous one.

The performance of these three forecast scenarios is evaluated with the Fiasco score which counts the cases when an above average (respectively below average) temperature is forecast and a below average (respectively above average) temperature is observed.

General comment

Coming from a slightly different field, some methodologies were unclear to me, and would require some clarifications. This particularly targets the Fiasco next score and the time series it is based on. Detailing the variables and thresholds considered in each Fiasco score (standard, detrended, next) could help in that matter.

Results are discussed in the Conclusion section. I would have liked more in-depth discussions of the results, either in the Results section or in a Discussion section, and a more pragmatic link between the results and the general objective of the paper: "help end-users to understand better how to use seasonal forecasts".

To which extent does the Fiasco score actually measure/reflect end-user needs? This score is very specific to worst-case scenarios.

Other general questions and comments:

- Section 2.2: Is it correct to say that you used the forecasts issued in May, June and July to obtain the forecasts for June-July-August at one month lead? If so, don't you have 1350 hindcast runs? If you do limit yourself to the first month lead, then this study would be an evaluation of monthly forecasts rather than seasonal forecasts. Lastly, which time step do you consider? From the rest of the article, it seems that the temperatures for June July and August are aggregated.

- The computation of the AUROC score needs to be better explained.

- L. 154-155 "However if both . . . 1981-2010 conditions." Just a comment: if these trends maintain in future years, couldn't the thresholds be adapted to allow for a fair evaluation of the models?

- Section 5 first §: Did I understand correctly: you change the variable of interest from simply being the temperature forecast to being the difference between the temperature forecast for one year and the previous one, thus resulting in 29 values instead of 30 values. Are the thresholds defining the three equal probability groups chosen within this sample of 29 values for each month? Within the sample of 29*3 values for all JJA months altogether? Additionally, if my understanding of this paragraph is correct and if all verification methods remain unchanged, the change in name "Fiasco next score" might be confusing as it would be the same score simply applied to a different variable.

- L.207 "The RPSS, which takes. . .is positive only in the tropics between 22S and 21N." Isn't the RPSS also positive for latitudes south of 47S and between 20N and 43N with some exceptions around 26N?

- L.208 ". . .the latitude average number of fiasco is 7%..." I could not find these values in Figure 6.

- Why not consider the time series of differences in forecast between one season and the previous, but calculated from the detrended forecasts?

- To which extent is the trend in temperatures responsible for the "good Fiasco scores" (Figure 5a) as compared to the Fiasco next and Fiasco detrended scores (Figure 5b and 5c)? To which extent do these results inform us on optimal strategies for end-users to detect "fiascos"?

- L.243 "the end-user should consider using the coming season forecast relative to previous season or a shorter reference period than the traditional 30 years. . .": I would have liked to see this already in this paper to strengthen the analysis.

Technical issues:

- Throughout the paper, spaces are missing between words or after punctuations and special characters. Extra parenthesis also appear, e.g. pages 7, 8 or 9. Several typos appear in the text. I listed some of them below.

[Figure]

- L.45 Change "observes" to "observed"

- L.48 Many other methods exist to evaluate hindcast skill. This sentence should not be restrictive to the criteria enumerated here.

- L.50 I would suggest to change for example to : "...the Area Under the Relative Operating Characteristic (AUROC) curve which considers jointly the hit rate (HR) and the false alarm rate (FAR), ..."

- L.51-52 Maybe add: "Epstein, E. S., 1969: A scoring system for probability forecasts of ranked categories. J. Appl. Meteor., 8, 985–987." When introducing the RPS

- L.69 could you detail the following: "their figures regarding their work"?

- L.80 "It became operational in November 2011"

- L.83 "one month lead"

- L. 131 "Spatial averaging of the model increases"

- Figure 2: Rewrite the legend to make (a) appear

- Figure 2: I could not see the dashed contour lines mentioned in the legend. Are they supposed to be seen in (a), (b) and (c)?

- L.146 "with the same scale as Figure 2a"

- L.161 Replace "leg" with "lag"?

- L.163 "hindcast"

- L.198 "radon" change to "random"

- L.210 change "significant" to "significantly"

---

## Author Comment (AC1) · 19 Feb 2017

Reply to Anonymous Referee #1

We appreciate the time and effort invested by the referee in reviewing this manuscript. We thank the reviewer for the helpful comments which we will be address to improve the manuscript, as highlighted below in blue:

To facilitate the assessment of risks, seasonal forecasts are usually expressed as the deviation relative to a 30-yr average condition for end-users. However, the climate and socio-economic changes during this period may enable the deviation questionable. To eliminate these changes, this paper introduces a method, which uses the previous year's season condition as a reference for the next season, to evaluate the skill and usefulness of the ECMWF system 4 (Sys4) seasonal June-July-August (JJA) temperature forecasts, wherein the ECMWF ERA-Interim reanalysis is employed as the surrogate observations. The results may be valuable for the policy-makers who are interested in the application of seasonal forecasts. The paper is organized logically and well writing. However, I still have a few concerns that need to be addressed.

We are grateful for these comments which confirms us in our concern about using seasonal forecasts.

Major comments:

1. Policy-makers or end-users may have great interests on the impacts of extreme event to the socio-economy, agriculture, and environment. In this paper, however, the metric of "Fiasco score" was constructed based on the above-normal and below-normal condition which does not simply equal to extreme event. The skill of Sys4 in terms of capture the extremes event should be added or discussed in the context.

One of the limitation of current seasonal forecast is that the forecast is given as an average value for 3 months. Obviously a situation with an extreme above normal episode (precipitation or high temperatures) can be followed by an extreme opposite situation causing the whole period to be "normal".

Although desired by end users currently there is no evidence that long range forecast can predict short term extreme events. In order to evaluate the skill for extreme events the daily or even hourly data are needed, while monthly averages are presented here.

This comment raises an interesting idea to check the Sys4 skill to forecast the most extreme season in the hindcast period (the coldest and warmest season). This can hopefully be performed.

2. As Figure 6 shows, the forecast skill derived by the new method (i.e., using the previous year's season as a reference for the next season forecast) tends to be lower than that derived from the traditional 30 year reference period. As well, the authors concluded that the new method is obviously not suggested to replace the robust traditional 30 year reference period (see Line 241-242). If so, why you attempt to use it?

The traditional 30 year reference period is the WMO guideline for climatologists. As a precaution measure we suggest to use both the 30 years climatology and the previous year as a reference for the next year forecast. However, there are relatively large areas over land that the new method over performs the 30 years reference period as indicated in Figure 6 and more visually in Figure 4. In order to be more decided we will add a case study to demonstrate the problem of using the 30 years climatology.

3. In my view, the precipitation or streamflow forecasts, as compared with temperature, may have more significance on end-users practice. Why you put the primary attention just on temperature, rather than precipitation?

Crop yield is also sensitive to temperatures and usually droughts come together with warm temperatures anomalies. However the main reason we choose the temperature field is the aim to check the conception that the temperature forecast skill is higher than precipitation. As shown in the paper a large source of temperature forecast skill originates from climate change.

Furthermore, the precipitation variance is usually larger than temperature and unlike temperatures which is usually distributed normally, precipitation may be right-skewned as there is a limit at zero and a long tail of high precipitation events. This may cause a situation where the seasonal precipitation value is below average but the tercile category will be above average. Therefore, the method for precipitation should be tested separately with even more caution.

Specific comments:

1. Section 2 – Model data. Please provide the archive website for ECMWF system 4 data and ERA-Interim reanalysis, respectively.

The ECMWF system 4 data is not open freely and only ECMWF members can retrieve the data. The ERA-Interim reanalysis is now open source so a link to the download site will be added.

2. Line 94-95. Please add an example table to explicitly illustrate the contingency table, so that the readers are more easily to understand it.

A new figure with a case study will hopefully illustrates the contingency table and the idea of the "Fiasco score".

3. Line 101-102. Please provide more details (e.g., equations) for the calculation of AUROC.

From the new case study that will be added also the AUROC will be calculated in order that readers can more easily understand the method.

---

## Author Comment (AC2) · 20 Feb 2017

Reply to Anonymous Referee #2

We appreciate the time and effort invested by the referee in reviewing this manuscript. We thank the reviewer for the helpful comments, from typos to fundamental correction, which we will address, as highlighted below in blue:

In a context of developing seasonal forecasts as climate services, this paper proposes an evaluation of seasonal temperature forecasts from ECMWF System 4 for the June-July-August season. Authors first propose the Fiasco score to evaluate the forecasts for risk assessment and cost-benefit analysis and relate it to the common RPSS and AUROC scores. They also present the trends in temperature from System 4 and ERAinterim.
They conclude that, with such trends, classifying forecasts between above normal, normal or below-normal conditions based on long-term thresholds will not be useful for end users as all future forecasts will be above-normal. If I understand correctly, the authors then analyze the System 4 temperature forecasts, along with two light post-processing: the detrended System 4 temperature forecasts and the series of differences in temperature forecasts between the current season and the previous one.

The performance of these three forecast scenarios is evaluated with the Fiasco score which counts the cases when an above average (respectively below average) temperature is forecast and a below average (respectively above average) temperature is observed.

We are grateful for these comment and we will improve the text by adopting the essence of the review.

General comment
Coming from a slightly different field, some methodologies were unclear to me, and would require some clarifications. This particularly targets the Fiasco next score and the time series it is based on. Detailing the variables and thresholds considered in each Fiasco score (standard, detrended, next) could help in that matter.

A case study will be added including a figure which illustrates the various fiasco scores. Some more explanation of the detrened and next score will be also added.

Results are discussed in the Conclusion section. I would have liked more in-depth discussions of the results, either in the Results section or in a Discussion section, and a more pragmatic link between the results and the general objective of the paper: "help end-users to understand better how to use seasonal forecasts".

The discussion will include some risk assessment and reference to the cost benefit ratio which may benefit the end users. A case study which will be added will serve as a warning to end-users regarding using of a seasonal forecast without understand the origin of the model skill.  The case study will emphasize the impact of climate trends on seasonal forecast skill.

To which extent does the Fiasco score actually measure/reflect end-user needs? This score is very specific to worst-case scenarios.

It is true that only the worst-case scenarios is addressed by the fiasco score. However given the low skill of seasonal forecast we believe that in the current situation it is the most important score for end user decision to trust or not to trust the seasonal forecast.

Other general questions and comments:
- Section 2.2: Is it correct to say that you used the forecasts issued in May, June and July to obtain the forecasts for June-July-August at one month lead? If so, don't you have 1350 hindcast runs? If you do limit yourself to the first month lead, then this study would be an evaluation of monthly forecasts rather than seasonal forecasts.
Lastly, which time step do you consider? From the rest of the article, it seems that the temperatures for June July and August are aggregated.

The forecast for all 3 JJA month is given in May, therefore it was an error to write a "one month lead". This is clarified now is section 2.1.

- The computation of the AUROC score needs to be better explained.

A case study will added to give an example so hopefully the AUROC will be better understood.

- L. 154-155 "However if both : : : 1981-2010 conditions." Just a comment: if these trends maintain in future years, couldn't the thresholds be adapted to allow for a fair evaluation of the models?

This is a simple solution that should be implemented for future forecasts. As this is a fundamental comment it will be added to the discussion:
"If using the 30 years climatology for future forecasts the average temperature trend of the forecast model should be added to the thresholds determining the tercile categories."

- Section 5 first §: Did I understand correctly: you change the variable of interest from simply being the temperature forecast to being the difference between the temperature forecast for one year and the previous one, thus resulting in 29 values instead of 30 values. Are the thresholds defining the three equal probability groups chosen within this sample of 29 values for each month? Within the sample of 29*3 values for all JJA months altogether? Additionally, if my understanding of this paragraph is correct and if all verification methods remain unchanged, the change in name "Fiasco next score" might be confusing as it would be the same score simply applied to a different variable.

You understood correctly the score. However the error using the term one month lead caused a problem in understanding the Fiasco next score. All the analysis was done for one season forecast from May aiming for the 3 month JJA. Therefore only 29 values remain and not 29*3. The term one month lead was removed.

- L.207 "The RPSS, which takes: : :is positive only in the tropics between 22S and 21N."
Isn't the RPSS also positive for latitudes south of 47S and between 20N and 43N with some exceptions around 26N?

- L.208 ": : :the latitude average number of fiasco is 7%..." I could not find these values in Figure 6.

Both comments are correct, the text refers to a figure 6 which contained the latitude average including sea and land (see figure added). In the current version only the average over land was presented. To clarify both images can be included in the manuscript. Defiantly the text will be changed accordingly.

- Why not consider the time series of differences in forecast between one season and the previous, but calculated from the detrended forecasts?

The use of the differences between one season and the previous causes the de-trending to be needless assuming the trend from one year to the other are negligible. Although not checked using the "Fiasco next score" on the de-trend data seems possible but will introduce both disadvantages of using 1 year as a reference and assuming a constant trend.

- To which extent is the trend in temperatures responsible for the "good Fiasco scores" (Figure 5a) as compared to the Fiasco next and Fiasco detrended scores (Figure 5b and 5c)? To which extent do these results inform us on optimal strategies for end-users to detect "fiascos"?

A correlation of 0.5 (r=0.5) was found between the temperature forecast skill and the temperature trend (by a skill score which is not presented). The correlation between the 3 scores and the temperature trend will be calculated and added.

- L.243 "the end-user should consider using the coming season forecast relative to previous season or a shorter reference period than the traditional 30 years: : :": I would have liked to see this already in this paper to strengthen the analysis.

As full analysis of different reference periods may be too long however this analysis for the added case study can be performed.

Technical issues:
- Throughout the paper, spaces are missing between words or after punctuations and special characters. Extra parenthesis also appear, e.g. pages 7, 8 or 9. Several typos appear in the text. I listed some of them below.

Hopefully all typos are corrected (it seems they originated from an old word version ...)

- L.45 Change "observes" to "observed"

Corrected

- L.48 Many other methods exist to evaluate hindcast skill. This sentence should not be restrictive to the criteria enumerated here.
Corrected

- L.50 I would suggest to change for example to : ": : :the Area Under the Relative Operating Characteristic (AUROC) curve which considers jointly the hit rate (HR) and the false alarm rate (FAR), : : :"
corrected, thanks

- L.51-52 Maybe add: "Epstein, E. S., 1969: A scoring system for probability forecasts of ranked categories. J. Appl. Meteor., 8, 985–987." When introducing the RPS
Although it is an old paper it is still relevant and therefore added, thanks.

- L.69 could you detail the following: "their figures regarding their work"?
Some examples were added.
- L.80 "It became operational in November 2011"
Corrected
- L.83 "one month lead"
Corrected and clarified
- L. 131 "Spatial averaging of the model increases"
The latitude average was performed on the various skills not the forecast itself. Therfore the skill should stay.
- Figure 2: Rewrite the legend to make (a) appear
Corrected
- Figure 2: I could not see the dashed contour lines mentioned in the legend. Are they supposed to be seen in (a), (b) and (c)?
The significant contours where removed from the figure but forgotten in the caption. Will be corrected.
- L.146 "with the same scale as Figure 2a"
Corrected
- L.161 Replace "leg" with "lag"?
Corrected
- L.163 "hindcast"
Corrected
- L.198 "radon" change to "random"
Corrected
- L.210 change "significant" to "significantly"
Corrected

---

## Author Comment (AC3) · 21 Feb 2017

The referee comment: The skill of Sys4 in terms of capture the extremes event should be added or discussed in the context.

The attached figure shows the percent of correct forecast of both maximum and minimum seasonal temperatures and only in case the maximum or minimum were correctly forecast.

It can be seen that in the tropics for less than 20% of grid points Sys4 managed forecast correctly both maximum and minimum seasons, and about 50% of the grid points the Sys4 managed to forecast correctly at least one of the extreme seasons. For higher latitudes the hit score the ability to forecast correctly both extreme seasons' approaches zero. However in mid latitudes (20-60) the Sys4 manages to forecast correctly at list

on extreme season it a rate above 20% which is encouraging.

[Figure]

**latitude average of extreme hit acore**

Fig. 1. Latitude average of extreme seasons forecast hit score